# Detection and Utilization of Reflections in LiDAR Scans through Plane Optimization and Plane SLAM

**DOI:** 10.3390/s24154794

**Published:** 2024-07-24

**Authors:** Yinjie Li , Xiting Zhao , Sören Schwertfeger

**Affiliations:** Key Laboratory of Intelligent Perception and Human-Machine Collaboration, ShanghaiTech University, Ministry of Education, Shanghai 201210, China; liyj2@alumni.shanghaitech.edu.cn (Y.L.); zhaoxt@shanghaitech.edu.cn (X.Z.)

**Keywords:** reflection detection, SLAM, plane optimization

## Abstract

In LiDAR sensing, glass, mirrors and other materials often cause inconsistent data readings from reflections. This causes problems in robotics and 3D reconstruction, especially with respect to localization, mapping and, thus, navigation. Extending our previous work, we construct a global, optimized map of reflective planes, in order to then classify all LiDAR readings at the end. For this, we optimize the reflective plane parameters of the plane detection of multiple scans. In a further method, we apply the reflective plane estimation in a plane SLAM algorithm, highlighting the applicability of our method for robotics. As our experiments will show, this approach provides superior classification accuracy compared to the single scan approach. The code and data for this work are available as open source online.

## 1. Introduction

Localization and mapping are key capabilities of mobile robots. Together with navigation, they often rely on utilizing LiDAR sensors to perceive the range of the surrounding environment [1]. Especially in urban and indoor scenes, plenty of reflective surfaces are present, which are prone to cause inconsistent range measurements because the beams of the LiDAR sensor may register the reflective surface, the reflected range, or, in the case of glass, the obstacle behind the glass. This can lead to errors in localization, mapping and navigation [2]; thus, it is preferable to classify and filter the sensor data accordingly.

Figure 1a shows an example of such a situation. This is a point cloud map generated from 187 individual LiDAR scans, featuring a total of 6,335,959 points. The collection was created inside the blue area, with windows and mirrors present. The points collected on the window/mirror are shown in green in this ground truth point cloud. All the orange points (696,138 in total) are reflected off the windows or mirror, while the red points are real obstacles observed by the sensor through the glass. This dataset also includes other reflected points in yellow. The dataset was generated by utilizing the ground truth texture depicted in Figure 1b.

As will be detailed later on, reflective surfaces and their reflections can be detected in LiDAR sensor data through specific patterns in the intensity, in combination with the incidence angle of the beam to the glass, as well as with LiDAR sensors that report more than one distance per beam (dual return). In our previous work [4] we reported on how to detect and utilize reflections from single LiDAR scans. Furthermore, we have investigated how deep point classification networks can be utilized to detect reflections, given our labeled training data [3]. Both of these approaches work on single LiDAR scans.

The core contribution of this paper is that we are building a map of reflective surfaces in the form of a set of reflection planes. Those planes feature optimized plane parameters and boundaries, as they are constructed from several observations in individual LiDAR scans. In the end, we use this global plane map of reflective surfaces to classify the LiDAR scans into normal points, reflective surface points, reflection points, and obstacle behind glass points (outdoor points in this study). Our work is targeted at indoor scenes, which usually contain many planar features. We specifically do not target small or non-planar reflective surfaces (e.g., metal legs of chairs or tables) since they are harder to detect and often do not have a surface that can be estimated easily. Windows and mirrors, on the other hand, have large planar surfaces, causing a large number of reflected beams. Therefore, we expect our plane approach to work well. Detecting that objects are on the other side of reflective planes is not easy, and we, thus, choose to remove both reflection points and obstacle behind glass points, keeping only points on the glass. This way, we only have points on real surfaces, which is good for SLAM.

We report two related approaches. In the first method, “Reflection Detection via Plane Optimization”, we utilize external localization, e.g., from a traditional Simultaneous Localization and Mapping (SLAM) approach, to build the plane map. In the second method dubbed “Reflection SLAM”, we implement our own, simple plane SLAM approach. The point of this approach is to show that the planes of the glass/mirrors can be utilized in a SLAM method. Our experiments will highlight the good performance of our approaches.

The key contributions of this paper are:Describing the theoretical foundations of reflection detection in 3D LiDAR scans;Developing a method for the construction of a global reflection plane map;Developing a method for Plane Simultaneous Localization and Mapping (SLAM) with normal and reflection planes;Highlighting the performance of the developed approaches in the experimental evaluation;Providing the code of our approaches as open source: https://github.com/STAR-Center/Reflection_detection, accessed on 20 July 2024.

This paper is structured as follows: In Section 2, we describe the related work. Reflections in LiDAR sensors will be introduced in Section 3 Sensor Modeling. An overview of the proposed system will be given in Section 4, while the details of both methods are introduced in Section 5. The experimental evaluation follows in Section 6, and the paper concludes with Section 7.

## 2. Related Work

### 2.1. Reflection Detection

There are many studies exploring the detection of reflective surfaces and removing reflections based on LiDAR data. The issue of LiDAR struggling to obtain accurate data for highly reflective optical objects was raised as early as 2004 [5]. Some of these studies focus on using 2D LiDAR input to locate specular surfaces, while others rely on multi-echo patterns or point cloud return intensity to identify potential reflective surfaces. Additionally, there are studies that utilize sensor fusion, integrating other sensors to collaboratively detect reflective surfaces.

Weerakoon et al. [6] utilize the sharp fluctuations in intensity values of laser data to identify glass or other transparent surfaces, and also integrate this algorithm result into the occupancy grid maintained by Cartographer [7]. Tibebu et al. [8] proposed an effective method for detecting and localizing glass using LiDAR, which is based on the variation of range measurement deviations between neighboring point clouds within a region. This approach employs a two-step filtering process, first detecting changes in the standard deviation of point clouds, and then further assessing the properties of the region based on changes in distance and intensity. The identification results are utilized to eliminate errors caused by glass in occupancy grid maps. However, both of these algorithms are based on 2D LiDAR data, and the generated results are occupancy grid maps.

Koch et al. [9] use a multi-echo scanner to record three returns pulses and analyze the mismatches in distance between scans. They use a Hokuyo 30LX-EW LiDAR to obtain point clouds and use two filters to discover potential reflective points and found decent results in 2D environments using TSD slam. They also try to use different values to distinguish the transparent and specular reflective material [10] and applying their method to 3D environments [11]. Zhao et al. [4] analyzed the interaction between laser beams and reflective surfaces, exploring the possibility of detecting reflective planes using methods based on dual return and intensity peak, and then classified point clouds based on the identified reflective planes. These papers demonstrate the advantages of multi-echo data analysis in detecting reflective surfaces.

Gao et al. [12] proposed a reflection noise removal method based on multiple LiDAR positions. They first convert point cloud data into range images with depth and intensity information, then utilize a sliding window to search for potential reflection areas on those images. Finally, they compare the candidate regions with neighboring frame data to filter out the reflection noise. Yun et al. [13] divide the space into surface patches and classify them into ordinary and glass patches based on the number of echo pulses corresponding to each region’s emitted laser pulses. Virtual points are detected and removed by examining the symmetry of reflections and geometric similarity. However, these papers all deal with terrestrial laser scanners, which are not widely used in fields such as mobile robotics.

Wu et al. [14] proposed integrating LiDAR with ultrasonic sensors to jointly identify reflections. This approach not only confirms the presence of reflective surfaces in front of the robot but also detects their specific location and size. Zhang et al. [15] proposed GD-SLAM based on the combination of a camera and a laser sensor. Firstly, it searches for regions of sudden intensity change in the point clouds, then utilizes camera data to generate gray-scale images of these regions to detect the presence of glass, and finally further updates the point cloud using the position of the glass. Yamaguchi et al. [16] propose a large-scale glass detection method based on the fusion of polarization cameras and laser sensors. Glass can only be detected by LiDAR at small incident angles, whereas at larger incident angles, the degree of polarization of light reflected from the glass surface increases. Their work combines the characteristics of both sensors to detect glass in the environment. Additionally, they improve mapping accuracy in reflective environments by calculating the degree of polarization from light intensity information [17].

Sonar sensors use sound waves to measure the distance to objects, but their data accuracy is not high, and their resolution is generally low. This makes it difficult to accurately detect the position and shape of reflective planes. Various cameras can use deep learning methods to identify glass areas in images, but they struggle to provide accurate distance information and are mainly used to supplement other sensing methods [18].

### 2.2. Plane in LiDAR-Based SLAM

In LiDAR SLAM, planes are one of the most common environmental features. Obtaining the position and geometric information of planes from laser data can be used to perceive the surrounding environment and build maps, providing crucial information for robot localization and navigation.

LOAM [19] uses curvature to classify points into edge points and plane points from the point clouds and solves the relative pose between scans by minimizing the distance from points to planes. They use three points in the target frame to determine a plane to calculate the distance from points to planes, but they do not recognize and record the entire plane.

Pathak et al. [20] studied the method of using mobile robots to achieve fast 3D mapping in predominantly planar environments. They proposed a pose registration algorithm that relies entirely on plane features. Compared to voxel-based methods, this approach determines the pose uncertainty by utilizing the uncertainty in plane features [21]. It is more robust and faster than other methods, and the plane features require less storage space.

BALM [22] introduces BA (Bundle Adjustment) LiDAR SLAM to reduce cumulative errors during the mapping process. BA relies on correspondences between feature points. Therefore, BALM uses an adaptive voxel data structure to find sets of plane points in each frame. Initially, the space is partitioned into voxels of 1 m units. Then, it is determined whether the points inside each voxel are on the same plane. If not, the voxel is divided into eight smaller voxels, and the process is repeated. BALM stores and associates planes in space using those voxels. However, this method divides a plane into many parts to represent it, which cannot provide a complete description of the parameters and boundaries of the same plane.

Favre et al. [23] proposed a plane-based point cloud registration approach for indoor environments inspired by the ICP algorithm. They introduced a method to find the best plane match based on plane characteristics. Additionally, they further optimized the relative pose between frames using a two-step minimization method, achieving robust results on real datasets.

The π-LSAM (LiDAR Smoothing and Mapping With Planes) proposed by Zhou et al. [24] is a real-time indoor LiDAR-based SLAM system that utilizes planes as landmarks and introduces plane adjustment as the backend of the system. They jointly optimize the parameters of the planes and the poses of keyframes. The algorithm maintains the local-to-global correspondence between points and planes scan by scan, accomplishing the recognition of planes in the point cloud and the computation of globally consistent poses.

## 3. Sensor Modeling

### 3.1. LiDAR Sensor

Light Detection And Ranging (LiDAR) is a technology that uses light to measure distances to objects. The LiDAR sensors used in our research and experiments are Velodyne HDL-32E and Hesai Pandar QT64. The former is a LiDAR widely used in the fields of SLAM and autonomous driving, with three methods for processing laser beam pulses: dual return mode, strongest return mode, and final return mode. In dual return mode, the sensor retains both the strongest and final return, while a return with insufficient intensity will be ignored. The Hesai Pandar QT64 is a short-range mechanical LiDAR that is used to record data in the dual return mode, which maintains the first and last return of the laser beam [25].

### 3.2. Reflection Model of Different Materials

Objects made of different materials have varying reflectivity and optical properties. Common indoor materials such as walls and wood primarily exhibit diffuse reflection to laser light, with minimal absorption and specular reflection, just like Figure 2a shows. These materials are ideal for the operation of laser ranging systems. Materials like glass and mirrors, which exhibit high reflectivity, differ significantly from ordinary materials. When interacting with a laser beam, mirrors primarily undergo specular reflection, as shown in Figure 2b, while glass, in addition to specular reflection, may also undergo transmission, with diffuse reflection occurring less frequently.

In [10], research has already been conducted on the optical interaction between laser beams and different materials. The influence of the angle of incidence on the interaction between laser beams and glass is presented in [26,27]. Figure 3a illustrates the reflection model of a laser beam interacting with glass. When the laser beam makes contact with the glass, there are three different reflection phenomena. Research on the relationship between the return intensity and the angle of incidence reveals that materials with reflective properties exhibit a peak in intensity when the laser beam approaches perpendicular incidence. As the angle of incidence decreases, the intensity rapidly decreases. As shown in Figure 3b, when the angle of incidence decreases to a certain degree (also related to the distance), the return intensity becomes too weak to be detected. At the same time, glass possesses transmittance, allowing some of the laser to pass through it. If there are other objects in its new direction, the light undergoes diffuse reflection and returns to the receiver along the same path. Despite the weakened intensity due to passing through the glass twice, the receiver may still be able to estimate the distance to the actual object. Glass also exhibits specular reflection, meaning that some of the light is reflected at the same angle of incidence. If the reflected light encounters an object in front of the glass, it will return to the sensor along the same path. However, since the sensor is unaware of this reflection phenomenon, it will mistakenly report the reflected point as being located on the other side of the glass.

Figure 3c illustrates a scenario where a laser beam from the Velodyne 3D LiDAR scanner is incident on a glass surface. In this scenario, the sensor will receive three peaks. The first return comes from the surface of the glass because it is at the closest distance. The second return comes from an obstacle on the same side of the glass as the sensor, and in this case, the intensity of this return is the strongest. The last received return comes from the laser beam transmitted through the glass, reflecting from an obstacle on the other side of the glass, and this is the last because the obstacle behind glass is at the farthest distance. Overall, in this scenario, the strongest return comes from the reflected point, while the last return comes from the obstacle behind the glass. The glass itself is ignored because it is neither the strongest nor the last return.

Based on the above model analysis, we can conclude that: 1. The return from glass can only be the nearest point, so in dual return mode, if the strongest return and the last return are different, the glass return can only appear in the strongest return, because the last return is obviously further away. 2. The intensity of the return from glass reaches a peak when the incident laser beam is perpendicular to the glass surface and decreases rapidly when the angle decreases. 3. Ignoring special conditions, such as fog or smoke, when the strongest and last return differs, it is quite likely that there exists a reflective surface in this direction—except for a few points that partially hit the edges of obstacles.

In addition to the scanner we used in our experiments, common scanners produced by other manufacturers also conform to the analysis above [4]. Based on the summarized patterns above, we have identified two highly effective methods for detecting reflective surfaces, which will be further explained through the following example analysis.

### 3.3. Intensity Peak Analysis

As analyzed above, in the multi-echo mode, the return from glass will only exist in the nearest return or the strongest return, not in the last return. Therefore, in the discussion of this method, only point clouds from the strongest return are used for analysis.

The colors from Figure 4a represent the intensity of the points—the closer the color is to red, the stronger the intensity is. The red box in this figure shows the area where the intensity peak occurs. In this scenario, the obstacles on the other side of the glass are far enough, so there are no returns from them. As a result, most of the glass is observable in the strongest return, and the intensity peaks occur at the center of it. In Figure 4b, we see the sensor’s height is not sufficient in this scenario, so there are no intensity peaks. These examples also illustrate that intensity peaks only occur when there is a laser beam from the sensor that is perpendicular to the reflective surface.

### 3.4. Dual Return Reflection Analysis

To emphasize the dual return analysis, we removed other point clouds from the scene, keeping only the four walls of the room, glass, wrongly reflected points, and objects on the other side of the window. As shown in Figure 4b, the point cloud is a combination of the strongest and last returns, with all points that are either in the last return or have a small difference between the strongest and last returns marked in blue. The remaining strongest return points are colored according to intensity, with green representing the weakest intensity and red the highest intensity. In this scene, due to the insufficient height of the sensor, there are no intensity peaks. Therefore, the points in the window section all have low intensity, displayed as green. Consequently, the intensity peak method does not work here.

Now we know that the returns can originate from three different sources: the glass itself, obstacles on the same side as the sensor, and obstacles on the opposite side of the glass.

We first consider the scenario where obstacles on the same side of the glass are closer to the glass than those on the opposite side. In this case, if the last return comes from the opposite side, the strongest return may come from any of the three situations. However, if the last return comes from the same side as the sensor, the strongest return may come from any situation except the opposite side. Additionally, when the last return comes from the glass itself, the strongest return can only come from the glass as well. So when the last and strongest are not the same, the strongest return should come from the glass or the obstacles from the same side, and the last return should come from obstacles from one of the two sides.

Secondly, if the obstacle on the other side of the glass is closer to the glass, through the same deduction, we can draw the same conclusion as above. That is, when the strongest and last returns are different in the same direction, the strongest return does not come from obstacles on the same side, and the last return does not come from the glass.

From the above analysis, we can conclude that returns from glass can only exist in the strongest returns. With this phenomenon, we can detect glass and attempt to obtain information about its surface. When we have the coefficients for the reflective surfaces and their boundaries, we may be able to use the obtained plane to classify the point cloud data, determine whether a point is affected by the reflective surface, and discern whether it is specular reflection or transmittance. Furthermore, we can even project the reflected points back to their original positions through mirror reflection to achieve better scan point utilization and observe objects beyond the sensor’s field of view.

## 4. System Overview

In this paper, based on the model phenomena and rules concluded above, we design a pipeline to process point clouds, searching for reflective planes within them. We use these planes to construct a global map consisting of reflective surfaces and then utilize it to reprocess each frame of point clouds. In our study, we classify point clouds similar to [3] but with some differences. Specifically, we classify points into the following classes: normal points, reflective surface points, reflection points, and obstacle behind glass points (outdoor points in this study).

We propose two pipelines, with the main difference being that the first one utilizes ground truth poses of the point clouds provided by the dataset for plane registration (e.g., coming from some traditional SLAM approach), while the latter employs a self-implemented plane-based SLAM algorithm to compute poses and simultaneously perform global plane optimization.

### 4.1. Reflection Detection via Plane Optimization

Figure 5a shows a diagram of the first pipeline, which performs the reflection detection via the plane optimization pipeline. The input data are a single laser scan from a LiDAR in dual return mode. In each iteration, the algorithm processes the strongest and last return to detect potential areas affected by reflections and attempt to fit reflective planes, acquiring their parameters and boundary information. Using the point cloud poses provided by the dataset, the planes are transformed into the world coordinate system. At this time, matching and plane parameter optimization operations are performed on the global plane map. Once all data are processed, an optimized global glass plane map is obtained. Subsequently, the obtained map is used to reprocess all laser scans. Leveraging the optimized plane information, particularly the refined boundary information, point cloud classification is processed for each frame, which will be superior to the original frame-by-frame classification, as our experiments will show.

### 4.2. Reflection SLAM

In Reflection SLAM, the algorithm did not use externally provided pose information for each frame. Instead, it employed a scan-to-map approach, utilizing environmental plane information for matching and calculating the pose of each frame. The specific steps are as follows: First, the algorithm finds potential reflection-affected points, as described above. These points are used to detect reflective planes. Additionally, they are removed from the original point cloud, which is then used to perform traditional plane extraction to build a local plane map. The reflective planes are then added to the local plane map, and subsequently, the algorithm registers the local planes with the planes from the previous frame to obtain the initial global pose of the current frame. The local planes are then transformed into the world coordinate system, and the current frame’s global pose is optimized using a scan-to-map approach. Then, the algorithm updates the global plane map based on the local planes (plane parameter optimization) and continues to perform the aforementioned operations on each laser scan to obtain the global plane map. Finally, the global plane map and the calculated pose for each frame are used to classify the point cloud data. The information on non-reflective planes in the plane map also aids in identifying reflected points more accurately in the classification step. Section 5 presents the blocks in the pipeline diagram in detail.

## 5. Method

### 5.1. LiDAR Scan Glass Detection

Based on the phenomena and rules described in Section 3, we design an algorithm to process a single laser scan and detect potential reflective surfaces. Unlike our previous work [4], we have changed the approach to finding boundaries.

#### 5.1.1. Process Data Packet and Convert to Organized Point Cloud

We use the Point Cloud Library (PCL) [28] to store, reorder and process the point cloud. We firstly separate the packets from LiDAR into the strongest and last return point sets. For now, these points are unorganized. An organized point cloud can be considered as a 2D matrix, where each row represents a ring of the laser scan, and each column represents a degree of the laser scan. We organized the point cloud according to the ring number and azimuth degree, which is calculated using the point location and the LiDAR parameter. In the following section, we can just determine that two points belong to the same beam (i.e., the strongest and last return) if their ring number and angle are the same.

#### 5.1.2. Intensity Peak

To identify the intensity peak regions in the laser scan, we select horizontal rings from the data and traverse all points within these rings using the organized point cloud. Initially, we check whether the intensity of points gradually increases from a low threshold of 30 to a maximum threshold of 70, and then decreases in the same manner. The thresholds mentioned come from the intensity values in the point cloud data within the PCL. Additionally, a maximum threshold for the distance between adjacent points is set to 0.3 m, as points on the same reflective plane should be close to each other. If the distance between adjacent points exceeds this threshold, the current sequence is considered invalid or the sequence is terminated at that point. This threshold is more like a qualitative parameter. Testing its feasibility during the development process can eliminate most of the non-compliant noise situations. Further validation is performed on all potential sequences identified within the horizontal ring. We select points at the same azimuth angle from two adjacent rings above and below and verify whether the intensity follows the same order in the vertical direction. If the validation is successful, the associated points in this region are retained, and an attempt is made to fit a plane using these points. If successful, these points will be considered as part of a reflective plane. This Algorithm 1 is described below.
**Algorithm 1.** Algorithm for finding intensity peak**Require:** 
cloud: organized strongest point cloud of one scan**Ensure:**

Points of every Intensity peak peakresult   Filter the point cloud to horizontal plane of the sensor(z axis) save to horizontalscan;   **for** each point in horizontalscan **do**        calculate gap between this point and last;        **if** gap>threshold **then**               **if** the point is in a sequence increasing to the max intensity and decreasing sequence to next gap **then**                    Add the point to the sequence        **else**               **if** the sequence is valid **then**                    Store the peak to potentialpeaks   **for** each peak in potentialpeaks **do**        Choose points from same azimuth degree (same column) in the upper and lower two rings        **if** Verify the points is also an increasing then decreasing sequence **then**        Store these points to peakresult

We use Random Sample Consensus (RANSAC) to fit a plane to the selected points using PCL. We set the minimum confidence and minimum inlier count as criteria to judge the success of the fitting in the segmentation model. Inliers of the plane, as well as the plane parameters, are stored.

#### 5.1.3. Dual Return

As described before in the dual return analysis, we can use the dual return method to search for reflections. Firstly, we utilize the organized point cloud extracted from dual return packets. We traverse all laser beam directions according to the rings and azimuths, checking if the distance between the strongest return and the last return belonging to the same laser beam is equal along that direction. If the distance is less than a threshold of 10 cm, we retain them. Otherwise, we consider that there exists a reflective surface in this direction. Most obstacles are not closer to the reflective plane, so in the process of development, we set 10 cm as the threshold, and it distinguishes the normal situation and the one with a reflective plane in place well. The point closer to the sensor between the two is regarded as potentially returned from the reflective surface. We use RANSAC to attempt plane fitting on these points, with certain requirements on the number of inliers for successful identification of the reflective plane. This Algorithm 2 is described below.
**Algorithm 2.** Algorithm of detect reflection using dual return**Require: **cloud: organized strongest point cloud of one scan and organized last point cloud of one scan**Ensure:**
Degree of glass and the fitted plane of glass   **for** each point in each ring (row of cloud) **do**         Calculate the distance of the strongest and last point in the same degree         **if** distance is the same **then**             Add to normalpoints         **else**             Save the point with lower distance and in strongest point cloud to glasspoints             Save rest points to remainpoints             Save the rings and degree to degreehasglass   **while** number of glasspoints>300 **do**         Fit plane using RANSAC         **if** plane has inliers more than 200 **then**             Save the fitted plane         Remove inliers from glasspoints

#### 5.1.4. Find Boundary

In our previous work, we assumed reflective planes to be rectangular, and for each plane detected through the fitting, the algorithm statistically computed the farthest glass point of that plane in each direction to compute the rectangular boundary. In contrast, in this paper, the convex hull algorithm is used to process the fitted inliers to obtain the boundary of the reflective plane. This approach is more reasonable, fitting the detected glass planes more accurately and assisting the algorithm in detecting reliable areas of reflective planes without being affected by their specific boundary shapes. Additionally, this approach is more suitable for matching planes across different frames and integrating them into the global plane map. In our implementation, we actually used the 3D convex hull approach from PCL as a readily available and fast implementation. After obtaining the 3D boundary points using the convex hull algorithm for the inliers, the algorithm projects these boundary points onto the plane according to the fitted plane parameters, and then stores both the plane parameters and boundary points.

### 5.2. Plane Map Update

After performing reflection detection on each laser scan and obtaining relevant data regarding the reflective planes, we aim to integrate them together to construct a global map of reflective planes. This map includes the parameters of each plane and the convex hull points representing their boundaries. To achieve this, the algorithm utilizes the pose obtained externally for each frame to transform the plane data from each frame into the world coordinate system. After transforming local planes, the algorithm matches and updates the planes frame by frame with the planes in the global map. Please refer to Section 5.7 for specific steps.

### 5.3. Global Point Classification via Plane Map

For both pipelines depicted in Figure 5, reflection detection via plane optimization and reflection SLAM, after obtaining the global reflective plane map, we are reprocessing each LiDAR point cloud to classify all points. The classes are normal points N, reflective surface points G, reflection points R and obstacle behind glass points O. According to the pose of each frame point cloud obtained externally, we transfer the current point cloud to the world coordinate system and obtain the coordinate of the laser beam emission starts. For each point in the point cloud, starting from the coordinates of the sensor, we calculate whether its emission direction intersects with the reflective plane in the global map and lies within its boundaries. Then, we calculate the distance between this point and the intersecting plane and determine its relationship with the reflective plane.

For a point, if the laser direction it presents does not intersect with any reflective plane in the map, it is preserved and considered in N. For points that might intersect with planes in the map, further evaluation will be conducted. The points in front of the reflective plane are N, the points close to the plane are G, and the rest of the points are behind planes, consisting of both R and O. Since we detected the reflective planes and obtained the spatial relationship already, it is easy to separate N and G, but it is not easy to distinguish R and O. Here, we propose a method to recognize whether a point is in R or O.

In the first step, we mirror the inside points N against the reflective plane and compare them with the points behind planes. Behind the plane points that are farther than the mirrored points are considered to be points returned from obstacles on the other side of the plane. Then, we mirror the other unclassified points back against the plane to the inside and trace the laser beams in the direction of the mirrored point to check if there is another point further, just like Figure 6a shows. If there are points farther away in that direction, it indicates that the mirrored projection is not valid. If the mirrored point is from an inside obstacle, then the sensor should not have received data farther in that direction. Therefore, the origin point of the mirrored point should come from obstacles on the other side of the plane. The next step is similar to the previous one, except now we have some points from the other side of the plane. Therefore, we can determine some points in R by tracing the laser beams in the direction of the outside points and checking if there are farther points in the same direction; this case is shown in Figure 6b.

After completing the above steps, there may still be some points left unclassified. For these remaining points, we mirror them indoors according to the corresponding reflective plane parameters. Then, within the classified set of N, we search for nearby indoor points. If such indoor points are found, we consider the origin points of the mirrored point to belong to R.

### 5.4. Reflection Back Mirroring

Suppose the reflection point cloud is R, the parameters of the reflective plane are described as ax+by+cz+d=0,d>0, where $N={\left(a,b,c\right)}^{T}$ is the normalized normal vector of this plane, and the mirrored point cloud is M. According to the Householder transformation, we can obtain the mirrored point cloud as follows:$$\begin{array}{c} M=R\cdot \left[\begin{array}{cc} I-2\cdot N\cdot N^{T} & -2d\cdot N\\ 0_{1\times 3} & 1 \end{array}\right] \end{array}$$
where *I* is the identity matrix.

### 5.5. Extraction of Ordinary Planes from Filtered Point Clouds

The first step in the reflection SLAM approach is to extract the planes from the point cloud. We perform plane extraction using the region-growing segmentation algorithm available in the Point Cloud Library (PCL).

This PCL region-growing process involves segmenting the point cloud by estimating normals for each point and grouping them into regions to ensure that points within each segment belong to the same plane as much as possible. In the approach of region-growing segmentation, points within a certain distance range, and if the difference between their normals is less than a certain threshold, are considered to belong to the same smooth plane and are thus grouped into the same segment. Subsequently, the Random Sample Consensus (RANSAC) method is applied to each segment to robustly fit a plane model, yielding the parameters of the planes. The algorithm attempts to fit planes starting from the segment with the most points and continues until a satisfactory plane cannot be fit.

This process is then repeated for the next segment. Currently, the algorithm extracts up to 25 planes from each frame. For each plane, we obtain the 3D convex hull of the plane inlier points, similar to the glass plane detection, to compute the 2D boundary polygon. For determining whether a plane is satisfactory, the algorithm considers factors such as the number of inliers and the size of the plane area. We discard planes with fewer than 200 points or an area of less than 0.3 square meters. However, when the point density is too low, we have higher requirements for the number of points. The point density is defined as the ratio of the number of inlier points to the area. If point density is lower than 100, we require the number of inlier points to be above 500.

Before starting the above algorithm, we attempt to fit a plane for all points with z-values less than 0, considering it as the ground plane of the indoor environment, and store it separately. This approach allows us to effectively identify and extract planar structures from the point cloud data, forming the foundation for further analysis and processing.

An example of the plane extraction process is shown in Figure 7; points belonging to the same segment have the same color, and red points are outliers. The convex hull boundary is shown using markers in RViz.

### 5.6. Plane Registration

After completing the plane extraction for each frame, the next step of our SLAM approach for planes involves matching the planes between adjacent frames, utilizing correct plane correspondences to compute the relative poses between frames, and completing the plane registration process. The plane sets processed in the following steps are a union of the set of ordinary planes extracted from the filtered point cloud, as described in the previous section, and the set of reflective planes. For registration, the only distinction between the two types of planes is that during plane matching, only planes of the same kind will be matched. This section introduces the core of a self-implemented plane-based SLAM algorithm that computes the poses for each frame of point clouds using both normal and reflective planes. We completed the entire framework based on a plain and simple plane-SLAM approach and as streamlined as possible, as there is currently no open-source SLAM algorithm that fully utilizes plane extraction.

Here is an overview of the plane registartion process:1.**Plane Matching**: We iterate through all plane pairs between two frames, and based on the proposed four parameters, we systematically assess the similarity between these pairs of planes. If all criteria meet the threshold, we consider this a matching candidate, which will proceed to the next step for further evaluation.2.**Mismatches Removal**: After obtaining potential plane matching candidates, we want to determine an initial transformation relationship between the two frames and utilize it to eliminate mismatches from the matching candidates. Specifically, we iterate through all combinations of plane pairs and count the number of inliers when using each combination to compute the relative pose. By selecting the model with the most inliers to be the optimal model, we remove mismatches using the selected model and use this model as an initial guess in the next step.3.**Pose optimization through Gauss–Newton Method**: After obtaining the filtered matches, we use the Gauss–Newton method to optimize the relative transformation between the two frames.

#### 5.6.1. Plane Matching

Since a maximum limit of 25 planes is set for the number of planes in each frame, we can simply iterate through all possible combinations of plane pairs between two frames and determine their relationship through similarity criteria. We conduct base plane matching on the following aspects derived from the extracted plane parameters:Plane Normal: For the plane normal, the distance from the origin (below), and the centroid coordinate (further below), we assume that consecutive scans are collected close to each other, such that the poses of matched planes should be similar.The normal of a plane represents its orientation and inclination, being a vector perpendicular to the plane. Similar planes should have similar normal vectors. The cosine value of the angle between two normal vectors can be computed using the dot product as follows:
(1)$$cos\left(\theta \right)=\frac{{\;\;}^{s}n\cdot^{t}n}{{∥}^{s}{n∥\cdot ∥}^{t}n∥}$$
where the subscripts *s* and *t* represent the source and target plane, respectively. As we aim for the two planes to be as parallel as possible, the computed cosine value should approach 1 as closely as possible.Distance from origin to plane: The distance from the origin point to the plane projected along the direction of the normal vector represents the distance from the origin to the plane. This difference in distance can be calculated as follows:
(2)$$distance_{origin}=\left|d_{s}-d_{t}\right|$$If the distance between the two planes along the direction of their normal vectors is close, it is considered that these two planes are very close and may be the same plane.Plane centroid coordinate: Because the point clouds from two consecutive frames are adjacent and the sensor poses are very close, the point clouds extracted from the two data sets for the same plane should be very close, including the average centroid coordinate of the inlier points:
(3)$$distance_{center}={∥point\_average_{source}-point\_average_{target}∥}^{2}$$If the difference between the distances of two planes is small, they are considered very close and may belong to the same plane; otherwise, they may be two different planes on the same infinite plane.Plane Area: The area of the plane is also considered a criterion for matching. Since the displacement of the sensor between two frames is very small, its observations of the same plane should be consistent. Therefore, we also require the difference in area between the matched planes to be small. The ratio of the area between two planes can be used as reference data for comparison.
(4)$$ratio_{area}=\frac{max(area_{s},area_{t})}{min(area_{s},area_{t})}$$Since the relative pose between the two frames is currently unknown, we cannot calculate the overlap area ratio of planes using boundary information. Therefore, when comparing between frames, we use the area ratio as reference data. If this ratio exceeds a certain threshold, it is considered that the two planes being compared do not belong to the same plane.

In the plane-matching process, the aforementioned similarity criteria are used to determine whether planes from two consecutive point clouds belong to the same global plane.

The thresholds set for plane normal, distance from the origin to the plane, plane centroid coordinate, and plane area are 0.8, 0.1, 0.5, and 1.5. The results from Equations (Equation 1) and (Equation 4) will be compared with the thresholds to determine if a match is good or not. The threshold is determined through gradual experiments to obtain the most suitable parameters during the development process. When all of these conditions are met, it is considered that the target plane matches the source plane, indicating a successful match.

#### 5.6.2. Plane Correspondence Registration

Next, we utilize the obtained plane matching correspondences to compute the 4×4 rigid transformation T, which best fits the source planes to the target planes. This transformation is defined as follows:$$T=\left(\begin{array}{cc} R & t\\ 0 & 1 \end{array}\right)$$
with *R* a 3×3 rotation matrix and *t* a 3×1 translation vector.

First, we present a closed-form solution for the plane registration problem. This derivation process is similar to that in [29], but excludes the part where matching points are used for computation. Let $\left\{\pi_{j}={{(}^{s}n_{j}^{T}{,}^{s}d_{j})}^{T}\right\}$ and $\{\pi_{j}={{(}^{t}n_{j}^{T}{,}^{t}d_{j})}^{T}\},j=1,...,N$ be corresponding planes set from the source and target point clouds, respectively.

The rotation and translation are decoupled, as studied in work [30,31]. The least squares solution for the rotation matrix can be obtained by minimizing:(5)$$\sum_{i=1}^{n}{∥R{\;\;}^{s}n_{i}-{\;\;}^{t}n_{i}∥}^{2}$$

This equation can be solved by using the Singular Value Decomposition (SVD) method. The specific process is as follows. First, we construct the covariance matrix *H*:(6)$$H=\sum_{i=1}^{n}{\;\;}^{s}n_{i}{{\;\;}^{t}n_{i}}^{T}$$
then we perform the SVD decomposition on *H*:$$H=U\Sigma V^{T}$$
and finally, deduce the optimal rotation matirx *R* from the SVD results:(7)$$\hat{{\;\;}^{t}R_{s}}=VU^{T}$$
where *U* is the left singular vectors matrix, Σ is the diagonal matrix of singular values, and *V* is the right singular vector matrix.

Next, we need to compute the translation vector $\pi_{t}$, which can be formulated as minimizing the following expression:(8)$$\sum_{i=1}^{n}{{∥}^{t}n_{i}^{T}{\;\;}^{t}t_{s}-{(}^{s}d_{i}{-}^{t}d_{i})∥}^{2}$$

The solution to this equation can be equivalently represented as the solution to the linear system $A^{t}t_{s}=b$ where
$$A=\left(\begin{array}{c} {\;\;}^{t}n_{1}^{T}\\ \vdots \\ {\;\;}^{t}n_{n}^{T} \end{array}\right),b=\left(\begin{array}{c} {\;\;}^{s}d_{1}{-}^{t}d_{1}\\ \vdots \\ {\;\;}^{s}d_{n}{-}^{t}d_{n} \end{array}\right)$$

The least squares solution for the translation vector can be expressed as $\hat{{\;\;}^{t}t_{s}}=A^{+}b$, where $A^{+}$ is the pseudo-inverse of *A*.

#### 5.6.3. Mismatch Removal

Through the above equations, we can calculate the rigid transformation between two point cloud frames using three pairs of non-parallel plane matches. However, the results of the plane matching step may still contain mismatches. Therefore, we adopt the idea of RANSAC to remove outliers in the matching results. As mentioned earlier, the ground plane is extracted and stored separately. To satisfy the constraint of three pairs of non-parallel planes, the ground match will always be one of the three pairs of planes. Then, we traverse all possible combinations, searching for the model with the highest number of inliers and using it to remove mismatches.

#### 5.6.4. Pose Optimization through Gauss–Newton Method

For better results, we utilize all plane matches after removing outliers. We derive the analytical solution of the rigid transformation between the two point clouds using the formulas mentioned earlier as the initial guess, and then optimize the transformation using the Gauss–Newton method. To formulate the problem into an optimization framework, we represent the rigid body transformation using a six-dimensional vector $t={\left(x,y,z,r_{x},r_{y},r_{z}\right)}^{T}$, where *r* represents the rotation angles around each coordinate axis. The residual for each plane match correspondence is defined as follows:(9)$$e\left(t\right)=\left(\begin{array}{c} n(t)-n\\ d(t)-d \end{array}\right)=\left(\begin{array}{c} {\;\;}^{t}R_{s}s^{n}-{\;\;}^{t}n\\ {\left[{{\;\;}^{t}R_{s}}^{s}n\right]}^{T}{\;\;}^{t}t_{s}+{\;\;}^{t}d-{\;\;}^{s}d \end{array}\right)$$

The problem of finding the optimal rigid body transformation t can be obtained by minimizing:(10)$$\hat{t}=\arg\min_{t}\sum_{i=1}^{n}{∥e\left(t\right)∥}^{2}$$

According to the Gauss–Newton method, we compute the first-order Taylor expansion of the residual formula:(11)$$e\left(t+\Delta t\right)=e\left(t\right)+J{\left(t\right)}^{T}\Delta t$$
where J(t) is the Jacobian matrix of the residual function in t. The original minimization problem has also been transformed into finding the variation that minimizes the new expression:(12)$$\hat{\Delta t}=\arg\min_{\Delta t}{∥e\left(t\right)+J{\left(t\right)}^{T}\Delta t∥}^{2}$$

Taking the derivative of the expression with respect to Δx and setting it to zero, we obtain the following linear system of equations with respect to Δx, known as the Gauss–Newton equation:(13)$$J\left(t\right)J^{T}\left(t\right)\Delta t=-J\left(t\right)e\left(t\right)$$
by solving this equation, we can obtain the increment Δt, and use it to iteratively update t until the increment is small enough. Finally, the optimal rigid body transformation is obtained.

### 5.7. Global Map Update

Through the plane registration between two consecutive point clouds, we can obtain an optimized rigid transformation between frames. Next, we can utilize this transformation along with the locally extracted planes from each frame to perform global mapping. This paper focuses on constructing a global plane map containing reflective planes. Therefore, the map does not retain specific point cloud data but instead preserves plane models, including their parameters and boundary data. We take the reference frame of the first point cloud as the world coordinate system. Using the computed inter-frame rigid transformation, we derive the initial pose of the sensor for each frame. Subsequently, we update the planes in the global map. The specific steps are as follows:1.**Global Localization and Coordinate Transformation**: Using the global pose of the previous frame and the rigid transformation between two frames, we derive the global pose of the current frame and employ this pose to transform the local planes of the frame into the global coordinate system. This facilitates subsequent operations for global map construction and updates.2.**Plane Matching**: The plane matching step is similar to the inter-frame matching discussed earlier, with the key difference being that we now have the initial global pose of the frame. This allows us to place the planes in the same coordinate system. Consequently, there is no need to compare the area difference of the two planes. Instead, we can use the convex hull boundaries to calculate the overlap area of the two planes. In practice, we project the boundary points of the planes in the current frame onto the global plane. Then, we compute the proportion of the overlap area between the two convex hulls. If the proportion exceeds a certain threshold, we consider it as a successful match. In our implementation, we take 0.7 as the threshold.3.**Plane Registration**: This step also follows the same procedure as described earlier. After filtering mismatches using the RANSAC-like method and optimizing the global pose of the current frame using the Gauss–Newton method, the local planes are re-transformed into the world coordinate system using the newly obtained global pose. Finally, the pose is saved for further use.4.**Plane Update**: After completing the plane registration, the successful matches of local planes are used to update the global planes, while the planes that were not successfully matched will be directly added to the global map. After each update of the planes, the map is inspected. If a plane has been on the map for a certain period, its observation count is checked. If the count is too low, the plane will be removed from the map. This approach effectively removes incorrectly extracted planes from the global map, as well as planes generated by dynamic objects.The step of updating global planes using successfully matched local planes can employ various optimization algorithms such as Gauss–Newton, Levenberg–Marquardt [32], or filtering methods. However, since this aspect is not the focus of this research, a simple parameter-weighted averaging method is chosen to update the global planes. Specifically, depending on the number of observations for each global plane, different proportions are used to perform weighted averaging of the parameters of the matched planes. This approach ensures map stability while promptly integrating new observational data. Regarding the convex hull boundaries of the planes, the convex hull boundary points of the local planes are projected onto the updated plane, and the convex hulls of both planes are merged to obtain new boundary information.

## 6. Experiments and Discussion

### 6.1. Datasets

We evaluate our method on the 3DRef dataset [3], which is a large-scale 3D multi-return LiDAR dataset designed for reflection detection. Three different LiDAR sensors, along with a camera, are mounted on a platform (Figure 8) and moving in indoor environments featuring various reflective surfaces. We specifically select data from the Hesai Pandar Qt64 Dual Return LiDAR for Experiments, as its height configuration results in more reflected points in its point clouds. Additionally, its ultra-wide vertical field of view of 104.2° and the availability of dual return data (first return and last return) make it suitable for our algorithm. The dual return feature is essential for this algorithm, and a larger field of view helps in obtaining more complete reflective plane boundaries, but it does not affect the practical applicability.

This dataset was collected in diverse indoor scenes, including office areas, corridors, and rooms with various reflective surfaces, including windows, mirrors and other reflective objects. Utilizing the generated global ground truth meshes, points in the LiDAR dataset are labeled based on their relationship with reflective surfaces. Specifically, over 20% of the point clouds are associated with reflective surfaces across the three dataset sequences. Table 1 provides label statistics for Hesai data, and a visualization example for sequence 1 is shown in Figure 1a. The dataset also provides benchmark evaluation, leveraging the PCSeg codebase and using three deep learning methods for point cloud segmentation. We compare the results of our algorithm with these benchmarks.

### 6.2. Evaluations

There are currently no open-source algorithm implementations for LiDAR reflection detection, so we leverage the trained models from 3DRef [3], including MinkowskiNet [33], Cylinder3D [34], and SPVCNN [35]. We chose the first two algorithms as our benchmark, which analyze 3D point clouds geometrically to identify reflective surfaces and reflections. However, since these models were trained on a subset of the 3DRef dataset, their results may be better than actual performance due to overfitting. Additionally, we will compare our new algorithm proposed in this work with our previous method, which only uses a single point cloud to complete reflection detection. We use the name plane optimized as the short name for reflection detection via plane optimization.

Since we are comparing segmentation algorithms, we will first use the standard precision/recall metrics for evaluation, as in Table 2. Recall is the fraction of correctly classified points out of the total number of ground truth points. Precision is the fraction of correctly classified points out of the total number of classified points by the algorithm. In Table 3, the indoor points represent the collection of normal points and reflective surface points (glass and mirror).

Furthermore, to investigate our objective of mitigating the effects of reflections, we conducted additional statistical analysis of the results. In the point cloud after removing those classified as reflection, we calculate the proportion of remaining reflection points and the actual removal rates of reflection points by each algorithm. It should be noted that the removal rates in the table do not always correspond to the recall value of the reflection. This is because during point cloud processing, algorithms may leave some points that cannot be specifically identified or classified, and therefore, removal operations are also applied to them.

### 6.3. PointCloud Classification Evaluation

We evaluated our algorithm on three sequences from the 3DRef dataset and compared the classification results with other methods. Since the dataset directly provides the ground truth label for each point, we can directly conduct statistical analysis on the classification results. It is worth noting that two deep learning methods originally distinguished mirrors from glass. However, during statistical analysis, we classified them uniformly as reflective surface points. Figure 9 shows part result from sequence 3 with the ground truth label and the result from the single frame-processed reflection detection and the reflection detection via the plane map. In Figure 9a, the reflection is marked as orange, while normal points, reflective surface points and obstacle behind glass are marked blue, green and red, respectively. To provide a clearer visualization, the point cloud of the ceiling portion has been set to invisible. The figure shows that the single frame-processed reflection detection method removes most of the reflection points but still keeps some in the point cloud. The plane-optimized algorithm removes almost all the reflection points.

Table 2 displays the classification precision/recall results for all methods. From the results, the plane-optimized algorithm exhibits the highest precision for normal points across all three different sequences. However, its normal recall is lower, which can be attributed to the fact that in the plane-fitting process, the algorithm can not separate different frames of reflective surfaces on the same infinite plane. This leads to the result that the points near the extracted planes and within the boundary range are classified as reflective surface points. Figure 10 shows an example of this kind of situation. This also explains why the glass/mirror precision value for the plane-optimized algorithm is low. On the other hand, Zhao’s [4] method, which only considers points with different distances between the strongest and final return as reflective surface points, achieves higher precision and less recall. However, the confusion between normal points and reflective surface points is insignificant, as the primary goal of this study is to remove reflection points that result in incorrect positions due to reflections.

Our algorithm exhibits low performance in recognizing reflection and obstacle behind glass points in this table, but this value actually can not represent the reflection-removing ability of the plane-optimized algorithm. These values are low, primarily due to the fact that both the reflection and obstacle are points on the opposite side of reflective surfaces, and the plane-optimized algorithm has difficulty in accurately classifying them only using spatial information. Especially when outdoor obstacles are close to reflective surfaces, we cannot differentiate them using the maximum indoor distance. However, the algorithm keeps most beyond glass points unclassified and will remove them from the point clouds, ensuring accuracy in the retained data.

Below we present the results that represent the performance of removing reflections. Non-reflection represents the points that are not classified as reflection, and indoor represents the union of points that are classified as normal points and reflective surface points. We suggest to remove obstacle behind glass points from the point cloud because they are not important for indoor SLAM and also may have more undetected Reflection points. This is why plane-optimized have a lower reflection removal rate but still have higher indoor precision in sequence 1 because the algorithm can remove almost all reflection points from the indoor points but still leave some in the obstacle behind the glass class. The non-reflection precision and indoor precision show that in the filtered point cloud, the output from the plane-optimized algorithm has the smallest proportion of reflection points in most cases. The reflection removal rate also shows that the plane-optimized algorithm removes most of the reflection points in all three sequences. Zhao’s [4] method has a lower reflection removal rate, which may be because it can not detect complete reflective surface boundaries from a single scan.

### 6.4. Reflection SLAM Experiment

We also test our reflection SLAM algorithm on the 3DRef dataset. However, due to some limitations of our simple planar SLAM algorithm in indoor environments, we only tested it on a portion of the data from sequence 3, consisting of approximately 1200 frames of point clouds. The specific limitations will be detailed at the end of this section. We use IoU to show the result for this experiment, which is calculated as the ratio of the intersection area to the union area between the classification and ground truth regions. A higher IoU indicates a better match between the predicted and ground truth regions. Table 4 shows the IoU result for the classification experiment, and Table 5 shows results related to reflection removal. The reflection SLAM algorithm achieves similar performance as the reflection detection via plane optimization without the given ground truth pose data. The glass/mirror IoU is worse than the reflection detection via plane optimization because the output pose of the reflection SLAM algorithm is not accurate enough and will affect the plane parameter accuracy, which results in worse classification results in reflective surface points and the reflection class.

There are still some issues with this algorithm. The calculation of relative transformation between two point clouds requires at least three pairs of non-parallel planes because solving for the pose involves minimizing Equations (Equation 5) and (Equation 8), and at least three pairs of parameters are needed to obtain a valid solution.

In cases like corridors, where sensors can only perceive planes in two directions, the ground and two parallel walls, the algorithm will yield poor results because it lacks one degree of freedom in the information. The limitation comes from the planar SLAM algorithm providing localization, which leads to the inability of reflection detection to function properly. However, the reflection detection algorithm itself can work normally given good localization, as shown in previous sections.

### 6.5. Mapping around the Corner

After classifying the point cloud using detected reflective planes, we can not only remove potential reflection points but also utilize them for meaningful purposes. As analyzed earlier, the positions of these reflection points are symmetric about the reflective planes. Therefore, we can mirror them back to their original positions. This not only provides additional data for mapping but may also help the robot perceive objects that were originally out of its field of view. Figure 11 shows an example of our experiments. The white cube indicates where the robot is located and the mirrored back reflection points are marked in red. The classified reflective plane points are marked as green, and classified indoor normal points are marked as black.

In this scenario, the robot’s position is in front of a door, which is the entrance of a room, and the sensor is facing toward the interior. On the wall in front of the robot, there is a reflective plane whose extracted boundary is shown as a pink marker. Utilizing the reflection from this surface, the robot is now able to perceive the wall that was originally out of its field of view.

## 7. Conclusions

We have developed a method for reflection detection using dual-return LiDAR. Reflection detection via plane optimization utilizes externally provided pose information to construct a global map of reflective planes and classifies point clouds based on plane parameters and boundary information, removing misaligned reflection points. Reflection SLAM can construct a global map of planes without external pose information, optimizing planes uniformly and providing pose and reflective plane information for the classification step.

Our experiments show that we can accurately remove most reflection points from point clouds, ensuring the accuracy of point positions. The globally optimized map of reflective planes using multi-frame information performs better than obtaining reflective planes from single-frame data alone. Reflection SLAM can also achieve similar results in applicable scenarios. By identifying reflection points and mirroring them back using plane information, we can also achieve `mapping around the corner’.

In future work, we hope to differentiate between different reflective surfaces on the same infinite plane to obtain better boundary information and improve the accuracy of point cloud classification, which we assume will help to better distinguish the reflection point from obstacle behind glass points, thus solving the problem when obstacle behind glass points belong to the indoor environment. Additionally, we will continue to optimize our reflection SLAM, combining it with point-matching algorithms to obtain more robust and accurate results in more scenes and exploring other joint optimization algorithms to obtain better planes.

## Figures and Tables

**Figure 1 sensors-24-04794-f001:**
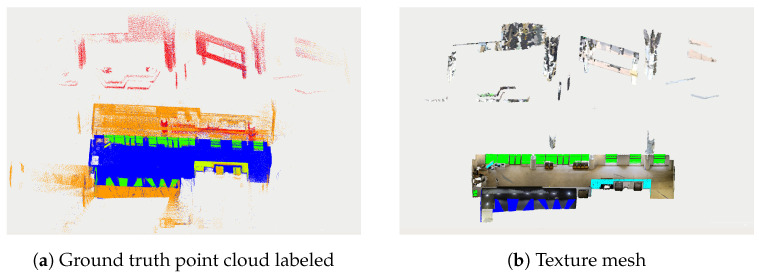
Reflective environment example [3].

**Figure 2 sensors-24-04794-f002:**
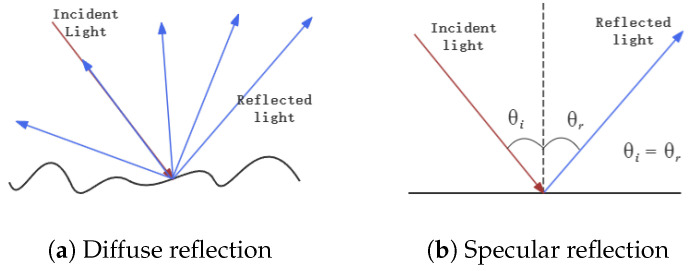
Laser beams interact with different materials.

**Figure 3 sensors-24-04794-f003:**
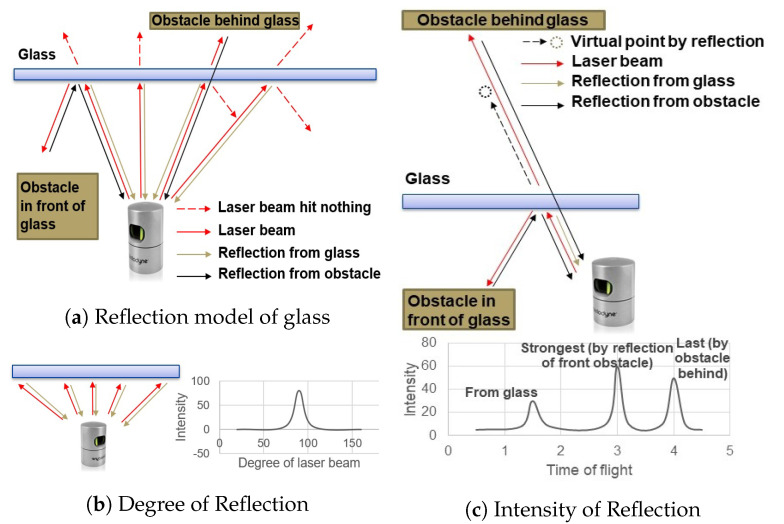
Laser Reflection Model [4].

**Figure 4 sensors-24-04794-f004:**
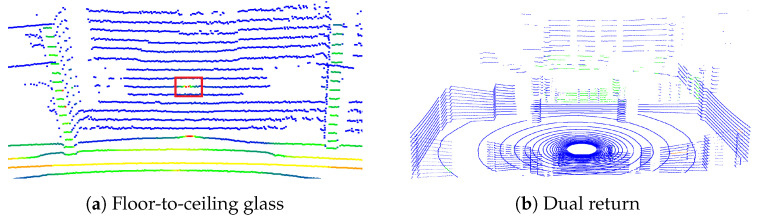
Point cloud examples [4]. The red rectangles indicate the areas where intensity peaks exist.

**Figure 5 sensors-24-04794-f005:**
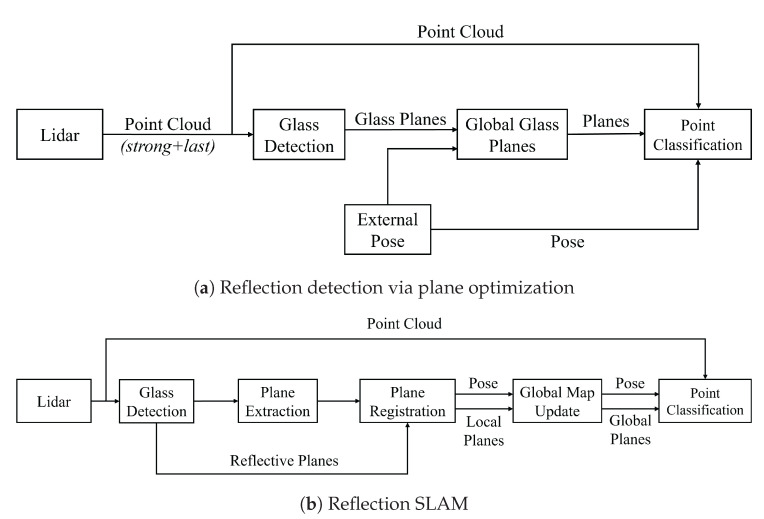
Approach overview.

**Figure 6 sensors-24-04794-f006:**
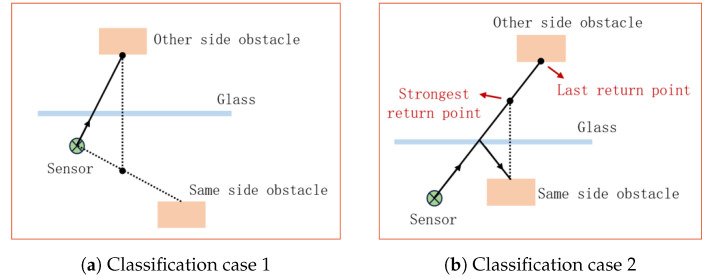
Classification instance diagram.

**Figure 7 sensors-24-04794-f007:**
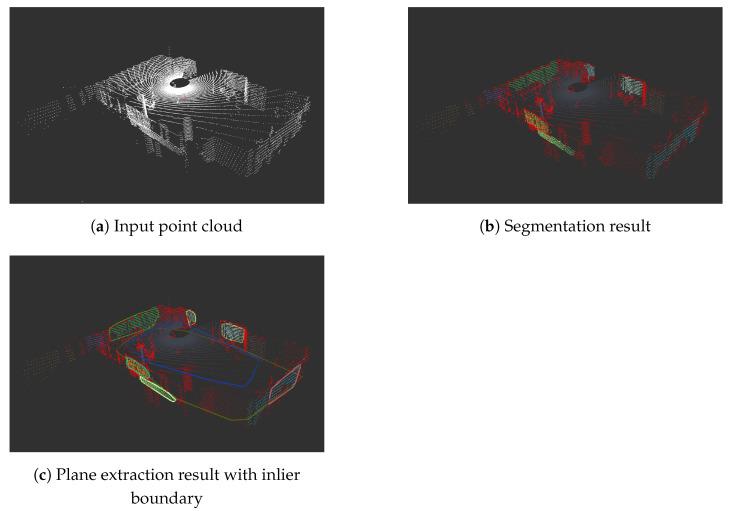
Plane segmentation using region-growing approach.

**Figure 8 sensors-24-04794-f008:**
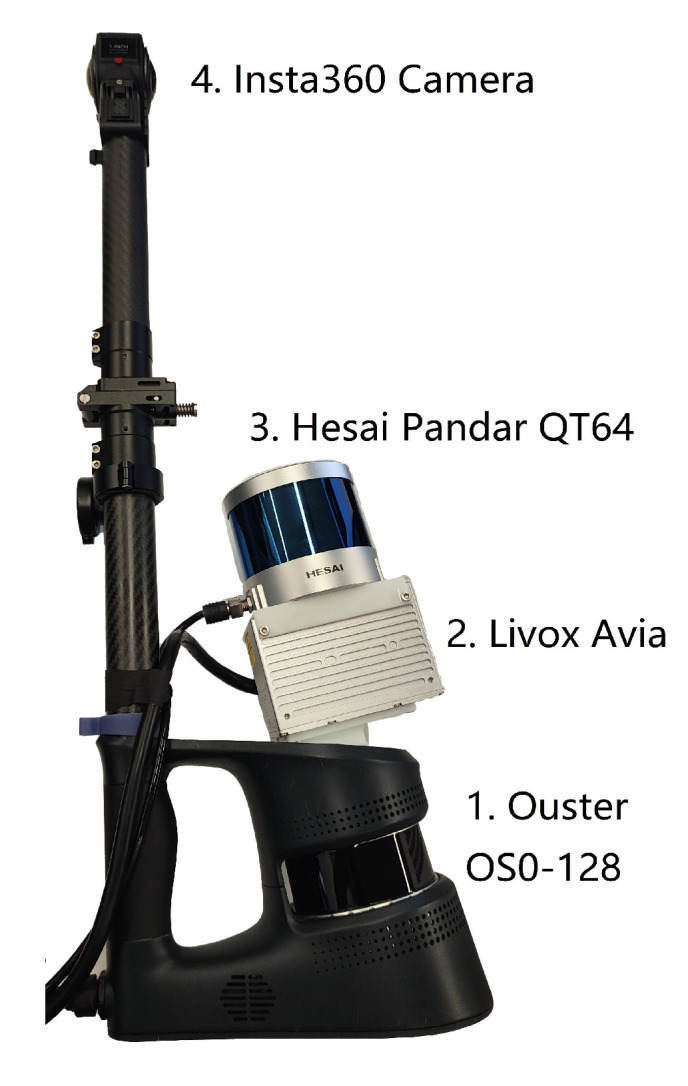
Data collection platform from 3DRef [3].

**Figure 9 sensors-24-04794-f009:**
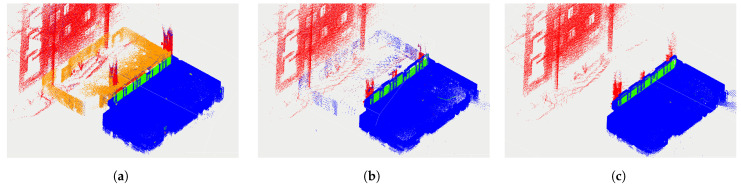
Visualization comparison of ground truth and results from reflection detection algorithm. (**a**) Ground truth labeled. (**b**) Zhao’s algorithm [4] labeled. (**c**) Plane-optimized algorithm labeled.

**Figure 10 sensors-24-04794-f010:**
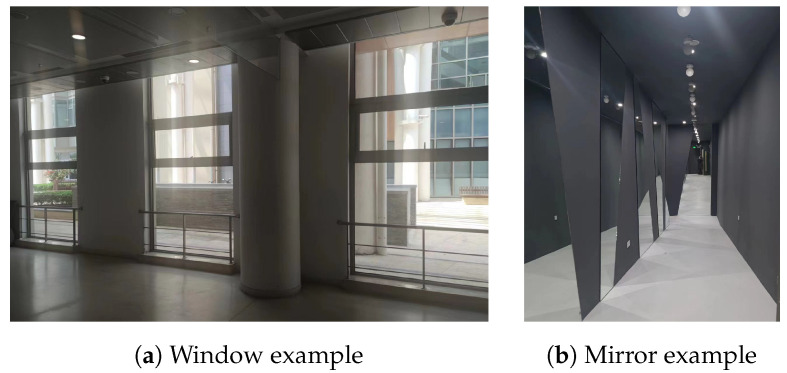
Example of a mixture of reflective surfaces and walls.

**Figure 11 sensors-24-04794-f011:**
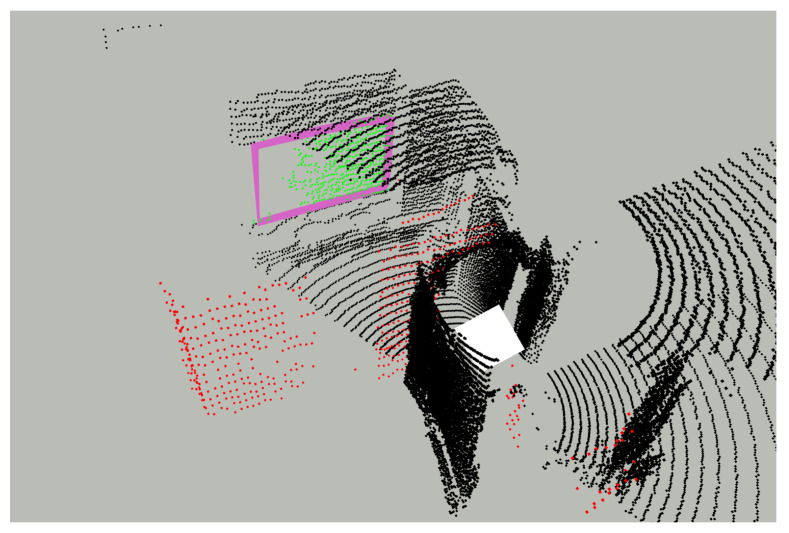
Classified result with reflection points mirrored.

**Table 1 sensors-24-04794-t001:** Percent of point label in each sequence for Hesai sensor.

Data/Size	Sensor	Normal	Glass	Mirror	OtherRef	Reflection	Obstacle
3Seq 1 3732	Ouster	84.19	0.91	0.61	2.85	9.31	0.66
	Hesai	76.67	3.16	3.41	2.60	10.77	1.54
	Livox	52.23	4.73	1.72	2.42	29.22	7.62
3Seq 2 4702	Ouster	87.75	2.59	0.04	3.23	2.41	2.45
	Hesai	78.64	7.48	0.28	3.14	3.79	4.16
	Livox	68.71	6.34	0.13	1.73	11.60	9.48
3Seq 3 7574	Ouster	87.05	2.17	/	2.76	1.67	3.64
	Hesai	76.97	7.76	/	2.14	3.41	5.46
	Livox	55.78	5.33	/	1.29	14.34	19.28

**Table 2 sensors-24-04794-t002:** Classification precision/recall results.

Methods	Sequence	Normal Precision/Recall	Glass/Mirror Precision/Recall	Reflection Precision/Recall	Outdoor Precision/Recall
Cylinder3D [34]	1	96.63/**98.56**	89.30/89.76	**93.90**/96.72	**89.94**/**91.87**
	2	94.91/**97.74**	**78.65**/80.92	**86.37**/82.43	**76.73**/**79.40**
	3	92.83/**98.53**	**90.97**/91.22	**89.08**/**92.22**	**91.40**/**91.54**
Minkowski [33]	1	97.46/98.23	88.40/**92.48**	91.99/**97.27**	86.88/89.42
	2	95.69/97.05	76.22/**84.50**	82.44/80.73	71.11/77.01
	3	94.06/98.08	90.68/**92.49**	85.25/89.74	84.00/90.60
Zhao’s [4]	1	93.51/89.31	**93.05**/60.01	84.96/1.87	31.37/38.03
	2	94.49/85.86	69.60/32.72	4.98/0.59	31.05/7.49
	3	93.79/87.07	90.46/48.26	38.57/1.16	61.43/25.01
Plane Optimized	1	**99.42**/90.23	55.48/82.91	74.65/69.97	72.54/38.52
	2	**98.91**/91.18	54.42/72.59	31.77/**84.17**	29.23/2.38
	3	**98.20**/88.68	57.18/76.61	60.09/68.78	89.85/32.76

Bold values indicate the best performance.

**Table 3 sensors-24-04794-t003:** Filtered point cloud precision and reflection removal rate.

Methods	Sequence	Non-Reflection Precision	Indoor Precision	Reflection Removal Rate
Cylinder3D [34]	1	99.62	97.22	96.72
	2	99.30	96.05	82.43
	3	99.72	93.87	92.21
Minkowski [33]	1	**99.66**	97.77	**97.26**
	2	99.24	96.36	80.07
	3	99.63	94.90	89.74
Zhao’s [4]	1	96.77	94.07	74.96
	2	99.01	95.61	66.19
	3	98.87	94.86	80.20
Plane Optimized	1	99.61	**99.82**	96.53
	2	**99.67**	**98.98**	**88.45**
	3	**99.83**	**98.36**	**92.63**

Bold values indicate the best performance.

**Table 4 sensors-24-04794-t004:** Classification result comparison with reflection SLAM (IoU).

Methods	Total	Normal	Glass/Mirror	Reflection	Obstacle
Cylinder3D [34]	**98.48**	99.01	**80.62**	**86.14**	**90.58**
Minkowski [33]	98.35	**99.12**	75.99	79.55	86.56
Zhao’s [4]	92.51	94.70	42.80	5.96	43.71
Plane Optimized	95.55	97.56	52.29	53.33	58.15
Reflection SLAM	95.42	97.30	50.29	52.03	61.21

Bold values indicate the best performance.

**Table 5 sensors-24-04794-t005:** Result comparison with reflection SLAM.

Methods	Non-Reflection Precision	Indoor Precision	Reflection Removal Rate
Cylinder3D [34]	99.72	99.63	92.21
Minkowski [33]	99.90	99.60	88.07
Zhao’s [4]	99.88	99.73	85.71
Plane Optimized	**99.94**	**99.96**	**93.33**
Reflection SLAM	99.81	99.94	93.26

Bold values indicate the best performance.

## Data Availability

The sensor datasets and ground truth data utilized in this work are available online http://3dref.github.io/, accessed on 20 July 2024 [3]. The source code of this paper is available here https://github.com/STAR-Center/Reflection_detection, accessed on 20 July 2024.

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
