# Peer review of "Detection and Utilization of Reflections in LiDAR Scans through Plane Optimization and Plane SLAM"

_sensors, 2024, doi:10.3390/s24154794_

Round 1

Reviewer 1 Report

Comments and Suggestions for Authors

1. This study is mainly applicable to scenes with obvious planar features, and the author needs to provide clear definitions.

 2. In section 5.5.1, ”two points are considered to be in the same direction if both their ring numbers and degree values are the same”, this statement is not rigorous. The above statement only holds true when the SLAM system platform collects data in the horizontal direction or moves only in the vertical direction; Or if the interval between adjacent frames is very short, in all directions, especially when the rotation is relatively small, the above assumption is true. When the data collection system has 6 directions of motion simultaneously and there is a certain time interval, the above assumption does not hold.

 3. In the manuscript, a large number of rules and assumptions were adopted. Although the author's meaning can be understood from the textual discussion, it is still recommended to appropriately configure diagrams or renderings to make the problem easier to understand.

 4. In the experimental analysis section, it is recommended to provide a rendering of a large scene and analyze it.

Author Response

Comment 1: This study is mainly applicable to scenes with obvious planar features, and the author needs to provide clear definitions. 

Response 1: Thank you for your comments, we think this is a very good point. We have now added description about the applicable scenarios in the introduction.

Comment 2: In section 5.5.1, ”two points are considered to be in the same direction if both their ring numbers and degree values are the same”, this statement is not rigorous. The above statement only holds true when the SLAM system platform collects data in the horizontal direction or moves only in the vertical direction; Or if the interval between adjacent frames is very short, in all directions, especially when the rotation is relatively small, the above assumption is true. When the data collection system has 6 directions of motion simultaneously and there is a certain time interval, the above assumption does not hold.

Response 2: Thank you for your suggestion. However, you may have misunderstood the scope of the sentence. In this context, we are referring to the dual return mode, where within a single frame, the LiDAR emits laser beams in one direction and obtains both the strongest and last return points. Their ring numbers and angles should be consistent. We have also revised the original sentence to reduce the possibility of misunderstanding.

Comment 3: In the manuscript, a large number of rules and assumptions were adopted. Although the author's meaning can be understood from the textual discussion, it is still recommended to appropriately configure diagrams or renderings to make the problem easier to understand.

Response 3:
Sorry, we don’t know to what exactly you are referring. We hope to have covered your concern with our edits of the images and text in the manustript.

Comment 4: In the experimental analysis section, it is recommended to provide a rendering of a large scene and analyze it.

Response 4: We did this part in Fig. 9 and section 6.3, we focus on comparing the results between the previously proposed single frame processed Reflection detection method and the Plane Optimized algorithm proposed in this paper. This dataset is a 250m^2 room - the biggest dataset available with ground truth information.

Reviewer 2 Report

Comments and Suggestions for Authors

The authors provided good work in the field.

The comparative assessments of the article are mainly compared with the results of paper [4]. Experimental results for detecting objects on the other side of the mirror are not good. (So ​​the authors chose an experimental environment where the other side of the reflective glass was an outdoor environment and argued that the research was conducted in an indoor environment so outside objects were not important). If it is an indoor environment but there are obstacles behind the mirror that the robot will pass through, it will be difficult to be effective.

However, the mathematical formula part is not closely related to the tests and parameters of the experiment. (in the form of theoretical foundation formula). The authors can either add or explain more about the results based on the equations provided the could bring the results from math problems to the experiments.

Comments on the Quality of English Language

The writing is good to follow.

Author Response

Comment 1: The authors provided good work in the field.
The comparative assessments of the article are mainly compared with the results of paper [4]. Experimental results for detecting objects on the other side of the mirror are not good. (So the authors chose an experimental environment where the other side of the reflective glass was an outdoor environment and argued that the research was conducted in an indoor environment so outside objects were not important). If it is an indoor environment but there are obstacles behind the mirror that the robot will pass through, it will be difficult to be effective.

Response 1:
 Thank you for your suggestion. We agree. It is very difficult to distinguish them. We have changed the text in the introduction and conclusion to make this issues more clear. 

Comment 2: However, the mathematical formula part is not closely related to the tests and parameters of the experiment. (in the form of theoretical foundation formula). The authors can either add or explain more about the results based on the equations provided the could bring the results from math problems to the experiments.

Response 2:

We agree that our experiment is not closely related to the mathematical formula, this is because that the first part, the construction and utilization of a plane reflectionp map, is the key part of the paper. While the planar SLAM algorithm part, which has many formulas, is mainly to show how reflective glass can be utilized in a SLAM method. We have added more explainations and analysis in the experiment part based on the equation, amd also emphasize the motivation for the planar SLAM algorithm in the text. Thank you for your suggestion.

Reviewer 3 Report

Comments and Suggestions for Authors

As backed-up by the experimental results presented in the paper (esp. table 3) the Plane Optimized algorithm yields good results when identifying and removing reflected points from the point cloud. This could be attributed to the fact that reflective surfaces in the 3dref dataset are “well-organized” as they come from mostly parallel-to-walls reflective surfaces. It would be of significant value to extend the research on datasets where reflective surfaces are placed differently. However, on this particular dataset the results are better than those of the previous works which is the main point of the paper.

The other issue with the paper is that the authors do not provide some of the parameters referenced in the text or used in the algorithm. Especially concerning is the elusive ‘threshold’ which is used several times in reference to different parameters. How was threshold selected? What is the final value of it (if there is one)?

Inclusion of reflective SLAM seems unnecessary as it provides worse results (solving a more difficult problem) and has not been compared with other SLAM methods. More results regarding parameter fine-tuning for Plane Optimized algorithm could be provided instead. Or inclusion of dynamic objects (authors mention that their plane update step should be capable of dealing with that, but there are no results to back it up).

Author Response

Comment 1: As backed-up by the experimental results presented in the paper (esp. table 3) the Plane Optimized algorithm yields good results when identifying and removing reflected points from the point cloud. This could be attributed to the fact that reflective surfaces in the 3dref dataset are “well-organized” as they come from mostly parallel-to-walls reflective surfaces. It would be of significant value to extend the research on datasets where reflective surfaces are placed differently. However, on this particular dataset the results are better than those of the previous works which is the main point of the paper.

Response 1: We agree that this work is only for big, planar reflective surfaces. Other reflective surfaces (such as metal table legs, etc.) are much harder to utilize and detect using our approach, since the reflection direction is not consistant and/ or their surface area is very small - current LiDAR and localization approaches just don't have the accuracy to map that precisely. We have added more info on that in the introduction.

Comment 2: The other issue with the paper is that the authors do not provide some of the parameters referenced in the text or used in the algorithm. Especially concerning is the elusive ‘threshold’ which is used several times in reference to different parameters. How was threshold selected? What is the final value of it (if there is one)?

Response 2: Thanks for pointing that out. We have provided the values and the reasons for the values in the paper.

Comment 3: Inclusion of reflective SLAM seems unnecessary as it provides worse results (solving a more difficult problem) and has not been compared with other SLAM methods. More results regarding parameter fine-tuning for Plane Optimized algorithm could be provided instead. Or inclusion of dynamic objects (authors mention that their plane update step should be capable of dealing with that, but there are no results to back it up).

Response 3: We agree that the first part, the construction and utilization of a plane reflectionp map, is the key part of the thesis. But we do think that using that map for plane SLAM is the logical next step. On the other hand, SLAM is a difficult task and modern methods use many tweaks and a multitude of different appraoches to achive good results - developing such a full blown SLAM in this paper is out of the scope - as is including dynamic objects. We have added more infos on the parameters.

Reviewer 4 Report

Comments and Suggestions for Authors

This manuscript presents a work which aims to classify reflections, reflective surfaces, normal points, and obstacle points located behind glass in LiDAR scans. Actually, the proposed Plane Optimized algorithm removes the reflection points in the 3DRef dataset and construct a global map of reflective planes. Detailed comments are as follows.

In the abstract, you mention enough background, please simplify it, and supplement necessary information about your work and its contribution.

Section 3.2,In addition to the scanner we used in our experiments, common scanners produced by other manufacturers also conform to the analysis above. There is no comparative analysis of different instruments in your study , it may be better to remove this sentence.

The content in Section 6.1 and Section 6.4 should be revised to better illustrate the practical applicability of the method.

Section 6.1, Additionally, its ultra-wide vertical field of view of 104.2° and the availability of dual return data (first return and last return) make it suitable for our algorithm.

Section 6.4, due to the limited applicability of the algorithm, we only tested it on a portion of the data from sequence 3.

Author Response

Comment 1:
This manuscript presents a work which aims to classify reflections, reflective surfaces, normal points, and obstacle points located behind glass in LiDAR scans. Actually, the proposed Plane Optimized algorithm removes the reflection points in the 3DRef dataset and construct a global map of reflective planes. Detailed comments are as follows.

In the abstract, you mention enough background, please simplify it, and supplement necessary information about your work and its contribution.

Response 1: 
Thank you for your suggestion. We have modified the abstract accordingly.

Comment 2:
Section 3.2,In addition to the scanner we used in our experiments, common scanners produced by other manufacturers also conform to the analysis above. There is no comparative analysis of different instruments in your study, it may be better to remove this sentence.

Response 2: We have cited the source for this statement. Additionally, we know this statement to be true, since it refers to the fact that modern LiDARs have dual return, which they do.

Comment 3:
The content in Section 6.1 and Section 6.4 should be revised to better illustrate the practical applicability of the method.

Section 6.1, Additionally, its ultra-wide vertical field of view of 104.2° and the availability of dual return data (first return and last return) make it suitable for our algorithm.

Section 6.4, due to the limited applicability of the algorithm, we only tested it on a portion of the data from sequence 3.

Response 3: Section 6.1 primarily describes the datasets used in our experiments. This sentence refers to the selection of three different LiDAR data types in the dataset. A larger field of view can help obtain more complete reflective plane boundaries, but it does not directly affect the practical applicability of the algorithm. We have now added some descriptions to clarify the scenarios in which the algorithm is applicable.

Section 6.4 primarily demonstrates the accuracy of reflection detection when the point cloud poses are provided by the planar SLAM algorithm implemented in this paper. The mentioned limitations mainly refer to the accuracy of planar SLAM in indoor environments, rather than the reflection detection algorithm itself. We have revised this section and indicated where the detailed description of these limitations can be found, which is at the end of this section.